# Not All Field Margins Are Equally Useful: Effects of the Vegetation Structure of Margins on Cereal Aphids and Their Natural Enemies

**DOI:** 10.3390/insects14020156

**Published:** 2023-02-03

**Authors:** Agnès Salat-Moltó, Berta Caballero-López, Nicolás Pérez-Hidalgo, José M. Michelena, Mar Ferrer Suay, Emilio Guerrieri, José M. Blanco-Moreno

**Affiliations:** 1Agroecology Group, Botany and Mycology Unit, Department of Evolutionary Biology, Ecology and Environmental Sciences, Faculty of Biology, Universitat de Barcelona, Av. Diagonal 643, 08028 Barcelona, Spain; 2Institut de Recerca de la Biodiversitat (IRBio), Faculty of Biology, Universitat de Barcelona, 08028 Barcelona, Spain; 3Department of Arthropods, Natural Sciences Museum of Barcelona, Castell Dels Tres Dragons, Av. Picasso, 08003 Barcelona, Spain; 4Institute for Integrative Systems Biology, C. Catedrático José Beltrán, University of Valencia CSIC, Paterna, 46980 Valencia, Spain; 5Department of Entomology and Pest Control, Cavanilles Institute of Biodiversity and Evolutionary Biology, University of Valencia, 46022 Valencia, Spain; 6Department of Zoology, Faculty of Biological Sciences, University of Valencia, Campus de Burjassot-Paterna, Dr. Moliner 50, Burjassot, 46100 Valencia, Spain; 7Institute for Sustainable Plant Protection, National Research Council of Italy, 10135 Turin, Italy; 8Department of Life Sciences, the Natural History Museum, London SW7 5BD, UK

**Keywords:** conservation biological control, semi-natural habitats, margin density, aphid–predator interaction, hedgerows, insect–plant interaction

## Abstract

**Simple Summary:**

Which plant life forms dominate in the field margin vegetation may affect its value in terms of the insect communities it harbors. We characterized marginal vegetation using the relative cover of each life form and sampled cereal aphids as well as some of their natural enemies (parasitoids, hoverflies and ladybugs) in crops along transects parallel to margins. Our results show that aphid abundance and parasitism rates were higher near margins dominated by perennial woody plants such as blackberries. By contrast, natural enemies had a clear preference for margins dominated by annual herbaceous species. By promoting specific life forms in already-existing margins, farmers can enhance the abundance of the natural enemies of aphids and decrease aphid pressure on their fields.

**Abstract:**

Differences in the semi-natural vegetation of field margins will affect the biological control services derived from the presence of these semi-natural habitats adjacent to fields. Of the plant functional traits that are most relevant for insects, plant life forms reflect different aspects of plant structure and functioning that can help predict the value of marginal vegetation for arthropods in agricultural systems. The aim of this study was to determine the effect of the vegetation structure of field margins on cereal aphids and on some of their natural enemies (parasitoids, hoverflies and ladybugs) in terms of plant life forms. We characterized margin vegetation using the relative cover of each life form and sampled insects in crops along transects parallel to field margins. Our results show that in the studied areas, the abundance of natural enemies was greater near margins dominated by annual plants than in margins dominated by perennial plants. On the other hand, the abundances of aphids and parasitism rates were higher near margins dominated by perennial woody plants than near margins dominated by perennial herbaceous plants. By promoting specific life forms in existing margins, farmers can enhance the conservation biological control and relieve aphid pressure on their crops.

## 1. Introduction

In recent years, research on biological control has devoted increasing attention to how semi-natural vegetation in agricultural matrix can guarantee the diversity and abundance of natural enemies of arthropod pests. This naturally occurring biological control is called conservation biological control [1,2] and its interest has grown over recent decades [3,4].

Research on the effects of semi-natural habitats on natural enemies and biological control has focused to date mainly on the correlation between natural enemies and the area occupied by these habitats [3]. However, semi-natural habitats in agricultural landscapes contain different plant communities that provide both natural enemies and pests with a variety of resources. Certain studies have explored the effects of different classes of semi-natural vegetation on natural enemies and pests based on the proportion of the area occupied by each habitat type in the landscape measured in buffers ranging from 250 m to several kilometers in width, e.g., [5]. However, the focus is seldom on the immediately adjacent field margin and its vegetation, even though it may be the nearest patch of alternative resources for organisms colonizing the crop [6]. In fact, field margins and their vegetation play a key role in modulating or enhancing conservation biological control [7,8]. In terms of management, the lack of attention devoted to field margins is surprising, since it is much easier for farmers to have an impact on these habitats adjacent to their crops than on habitat patches at larger landscape scales.

In recent years, many studies have recognized the importance of plant functional traits when attempting to disentangle the relationship between vegetation and insect biology [9,10,11,12]. An analysis of functional traits allows for comparisons between vegetation samples even if their species compositions differ [13,14]. Furthermore, instead of relying simply on vegetation species richness, which is often a poor surrogate for the breadth of niche opportunities offered by vegetation, the use of functional traits allows us to infer the resources provided by plants. From the many aspects relevant for insects, the life form integrates different characters of plant structure and functioning since it is a good indicator of both the height and the structure of vegetation, and whether or not the margin is permanently vegetated throughout the year [15,16]. Therefore, this information can help determine the value of margin vegetation for pest species, and for the conservation and promotion of biological control agents [7,8], since each group of organisms has its own behavior, life-cycle requirements and degrees of mobility that depend on traits such as body size and trophic level [17,18]. Other relevant functional traits are those related to floral resources, since predators and parasitoids also feed on nectar or pollen and a wide body of research has pointed out that floral resources significantly increase predator and parasitoid longevity [12,19,20]. Therefore, the availability of floral resources in time and space should also be taken into account.

Measures related to conservation biological control differ depending on the nature of different crops, their specific pests and their natural enemies. Cereal crops cover 30% of the continent’s total agricultural area (c. 53 out of 179 million ha; Eurostat, 2020) and so their management has a huge impact on the environment. The main arthropod pests in cereal crops are aphids, which are responsible for a 10–13% loss in wheat yields in the UK [21,22] and a 20% loss in barley yield in France [22].

Our study focuses on cereal aphids and three of their most important natural enemies: parasitoids (Hymenoptera: S.O. Parasitica), hoverflies (Diptera: Syrphidae) and ladybugs (Coleoptera: Coccinellidae). The aim of this study was thus to clarify the effects of the structure of margin vegetation on these groups.

In this study, we addressed the following three questions: (a) Does the vegetation structure of field margins measured in terms of the proportions of different plant life forms correlate with the abundance of the natural enemies of aphids? (b) Do different natural enemies of aphids respond differently to the structure of the field margin vegetation? (c) Do aphids and their natural enemies respond in similar or different ways?

Our hypothesis is that margins with simpler vertical structure (predominantly herbaceous) or with a discontinuous presence of standing vegetation throughout the year (mainly annual plants) will give rise to lower abundances of natural enemies in adjacent crops. More specifically, we hypothesize that the presence of ladybugs in cereal crops during the growth season will not depend on the margin vegetation since they are able to feed on aphids in the crops themselves in both their larval and adult forms. However, since the adults of parasitoids and hoverflies need sugars (and, in particular, floral nectar, which is scarce in cereal crops), we expect these groups of insects to be affected by the composition of margin vegetation as they may have to switch from crops to adjacent semi-natural habitats.

## 2. Methods

### 2.1. Sampling Site and Field Selection

Sampling was carried out in five areas in Catalonia (NE Spain) where cereals and legumes are the main crops, with autumn or winter cereals being selected. Field margins with semi-natural vegetation are present in all areas. The margin vegetation of these agricultural landscapes exhibits various degrees of complexity, ranging from annual flower-rich ruderal vegetation and perennial graminoid grasslands to scrubland and small copses. Field margins in these areas are not generally managed and, if they are, management is never intensive [23,24]. The selected margins were not managed directly during our study since we found no evidence of direct herbicide application or any controlled burning.

We used data from two partially overlapping datasets to test our hypotheses: one dataset consisted of data collected in one area (Gallecs) between 2014–2016, while the other dataset contains data collected in 2015 in Gallecs plus four other areas in central Catalonia (Figure 1). The sampling effort differed depending on the area and the year due to the different size of the study areas. In Gallecs (Ga), we sampled 39 cereal (wheat or barley) fields in 2014, 2015 and 2016, from the last week of April to mid-June. In each of the four remaining sites (Ca, Co, Es and Mo), we sampled four cereal fields (wheat or barley) in spring 2015, also from the last week of April to mid-June. These data sets enabled us to analyze the consistency of the response to the margin vegetation over time in one particular location (Gallecs; henceforth ‘time’ dataset) and spatially over one year (henceforth ‘space’ dataset).

### 2.2. Aphid Sampling

In each of the selected fields, we sampled aphids along a 50-m transect parallel to the field margin. Each ‘aphid transect’ was subdivided into five segments of 10 m each. In 2014, we sampled all five segments of each transect but were obliged to reduce the sampling effort in 2015 and 2016 to ensure a manageable number of samples. To guarantee a constant transect length, in 2015 and 2016 we sampled only the two end segments plus the central segment (Figure 2).

Each year, once a week, from late April until harvest time (usually mid-June), we collected ten cereal tillers as regularly as possible along the 10-m segments in each of the selected fields. Collected stems were carefully placed inside sealable plastic bags and kept in a portable fridge in the field. Afterwards the bags were kept at 4 °C for a maximum of four days until examination to ensure that the activity of any accidentally captured predator was low and thus avoid the loss of aphids.

On subsequent days, we removed all aphids from the plant stems using wet brushes and preserved them in 70% ethanol. Mummies from which parasitoids had not yet emerged were kept separately in individual vials covered with hydrophilic cotton at room temperature (20–24 °C, with no control over humidity) until the parasitoid emerged; both the parasitoid imagoes and the aphid mummies were then preserved in 70% ethanol. All aphids (including mummies) and hatched parasitoids and most hyperparasitoids were identified by species level if possible or otherwise by genus level. The aphids were identified by Nicolás Pérez Hidalgo, the parasitoids (Braconidae: Aphidiinae) by José M. Michelena Saval and the hyperparasitoids (Hymenoptera) by Mar Ferrer Suay (Cynipoidea: Figitidae: Charipinae), Emilio Guerrieri (Chalcidoidea: Encyrtidae) and Agnès Salat Moltó (Chalcidoidea: Pteromalidae, Aphelinidae; Ceraphronoidea: Megaspilidae) using specific bibliography for each group [25,26,27,28,29,30,31,32].

### 2.3. Predator Survey

In May of 2016, we undertook three surveys of hoverflies and ladybugs in six of the fields from which we had already sampled aphids (Figure 2). At the same site as the aphid-sampling transects, we conducted concurrent natural enemy visual transects at 3 and 20 m from the field margin between 10:00 and 19:00 h during fair weather (no rain, light wind and a minimum temperature of 17 °C). We counted adult, larvae and chrysalides of ladybugs, and larvae and chrysalides of hoverflies. We did not count adult hoverflies since their presence does not imply predation on aphids.

### 2.4. Vegetation Survey and Characterization

We established a 70-m transect centered on the aphid-sampling transect along the margin adjacent to the field edge. This ‘vegetation transect’ was divided into 10-m segments, in which we visually assessed the total vegetation cover and the relative cover per species using the Braun-Blanquet scale. To ensure standardized vegetation surveys, we only considered the vegetation in the 1-m-wide strip nearest to the field edge.

### 2.5. Vegetation and Data Analysis

Cover data were transformed into numeric percentages as per the Braun-Blanquet equivalences [33]. Given that it makes little sense to directly associate the information from vegetation transects with insect transects at segment level, we derived composite vegetation samples using a moving average of species-specific cover. Each composite sample represented the average of the vegetation cover in a block consisting of the three segments closest to each segment of the insect surveys, with each block partially overlapping with the next one. As a result, each segment in the insect transect corresponded to with one of the vegetation-averaged blocks (Figure 2).

We compiled the Raunkiær life form for all recorded plant species (therophytes: annual plants; geophytes: plants that have bulbs or rhizomes; hemicryptophytes: plants with buds at soil level; chamaephytes: plants with perennial shoots and resting buds near soil level (<25 cm); phanerophytes: shrubs and trees). We also compiled information on flowering attributes—specifically, on entomophily (the need for interaction with pollinators) and time of flowering (to check whether the species should have been flowering during our sampling)—as these traits affect the activity of natural enemies [4]. We used publicly available online databases [34,35,36,37] and a local flora [38] and crosschecked for inconsistencies between sources. Whenever the references did not agree, we prioritized data collected in Mediterranean areas. We used package TR8 version 0.9.18 [39] to access online databases.

We used the community weighted means (CWM) of the different life forms to characterize the margin vegetation. The CWM is the proportional cover attributable to each of the life forms present in each sample, and was computed using R Core Team 3.4.4 [40] with package FD version 1.0–12 [41,42]. To avoid collinearity between measures, we characterized the vegetation in each block using non-linear multi-dimensional scaling techniques (NMDS), retaining only two axes that represented the margin vegetation structure with a minimal deformation in reduced space (stress = 0.088). We used the package vegan version 2.5.2 [43] for multi-dimensional analyses.

Even though we initially intended to include entomophily in our models, we detected significant correlation with the first component of the vegetation NMDS (Spearman’s correlation coefficient: −0.374, *p* < 0.001; see Figure 3B). Entomophily was therefore excluded from linear models and only used to explain our results.

### 2.6. Insect Data Analyses

We summed the number of aphids, mummified aphids, hoverflies and ladybugs per transect over the whole sampling period to obtain the response variables. Parasitism and aphid, ladybug and hoverfly abundances were analyzed using different datasets. The sources of variation in aphid abundances and parasitism rates were tested with two independent datasets ‘time’ and ‘space’ datasets. Ladybug and hoverfly abundance, in turn, were assessed using the dataset ‘2016 visual transects’.

Both axes of the vegetation NMDS were used as fixed-effect covariates in generalized linear mixed effect models to test their effects on the response variables. After a visual inspection of the data, we detected lack of linearity in the relationship between response variables and the NMDS axes. Therefore, we included both linear (axis 1 and axis 2) and quadratic terms (axis 1^2^ and axis 2^2^) as covariates [44].

We also included crop variety as a fixed factor in the aphid abundance models. We employed a negative binomial distribution due to the overdispersion detected during data exploration. For the parasitism rates, models included aphid abundances as an additional covariate since parasitism rates can be density-dependent [45]. We used a logit-link function and a binomial distribution, appropriate for proportional data [44]. We used year as a covariate in models with the ‘time’ dataset and area in models with the ‘space’ dataset to account for any variation due to differences between years (owing to weather conditions) or areas (owing to regional differences in the aphid abundance or parasitism rate). For the natural enemies, models also included aphid abundances and distance to the margin as additional covariates. We used a Poisson distribution for the hoverfly model. A negative binomial was needed for the ladybug model to account for the overdispersion of data. Details of each model are summarized in Table 1. All models include the identity of the field as a random effect factor since observations (transects and segments) from the same field cannot be considered independent.

We checked the residuals graphically to ensure statistical assumptions were met [44]. We detected five outlier observations belonging to the same field and year in which aphid abundances were unusually high (up to ten times greater than the mean of all other fields). These points seriously affected our results even after modelling for overdispersion and therefore this field was removed from the analyses. All analyses were performed using R 3.4.4 [40] with packages glmmADMB version 0.8.3.3 [46,47] and lme4 version 1.1–17 [48] for model fitting.

**Table 1 insects-14-00156-t001:** Outline of the models fitted to each response variable, with an indication of the fixed-effect and random-effect variables included, and the distribution considered.

Dataset	Response	Fixed Effects			Random Effects	Distribution
‘time’ dataset	Aphid abundance	vegetation NMDS axis 1 + axis 1^2^ + axis 2 + axis 2^2^	+ crop	+ year	+ (intercept|field)	Negative binomial
‘space’ dataset				+ area		
‘time’ dataset	Parasitism		+ log(aphid abundance)	+ year		Binomial
‘space’ dataset				+ area		
2016 visual transects	Ladybug abundance		+ log(aphid abundance) + distance to margin			negative binomial
	Hoverfly abundance					Poisson

## 3. Results

### 3.1. Field Margin Vegetation

Margin vegetation can be characterized along two axes in terms of the predominant life forms. The first axis distinguishes margins dominated by annual or perennial vegetation (Figure 3A). The therophytes dominating the annual margins are mainly ruderal, short-lived plant species such as the common poppy (*Papaver rhoeas* L.), common fumitory (*Fumaria officinalis* L.), common sowthistle (*Sonchus oleraceous* L.) and white rocket (*Diplotaxis erucoides* (L.) DC.), most of which dry up in summer; on the other hand, margins dominated by perennial life forms are vegetated throughout the year. Furthermore, as stated above (see Vegetation analysis), we found that margins with low values on the first axis had more entomophilous plants in bloom during sampling (Figure 3B).

The second axis distinguishes between predominantly woody and herbaceous vegetation in margins dominated by perennial vegetation, which determines the vertical structure available during the year: in margins dominated by phanerophytes (mostly woody plants), a well-developed vertical structure persists throughout the year, while in margins dominated by hemicryptophytes they are more variable in the extent of green biomass over the summer, but never have a persistent vertical structure, and are almost never photosynthetically active all year round. Values around the center correspond to a greater diversity of life forms. Unlike hemicryptophytes, phanerophytes and therophytes, whose percentages of cover vary considerably between sampled margins (from 0% to almost 100%), chamaephytes and geophytes have little discriminant power since these forms only have a small range of plant cover.

### 3.2. Insects

We captured 51,623 aphids belonging to 14 aphid species, of which 4556 were parasitized; the most abundant species was *Sitobion* (*Sitobion*) *avenae* (Fabricius) (see Appendix A for a complete list of all aphid species found). We found 2897 unhatched mummies that we reared in the laboratory, from which we obtained 1020 primary and secondary parasitoids belonging to four families, twelve genera and twenty-four species. The most abundant species of primary parasitoid was the *Aphidius uzbekistanicus* Luzhetzki (Braconidae: Aphidiinae), while that of hyperparasitoids was the *Pachyneuron aphidis* (Bouché) (Pteromalidae) (see Appendix A for more details of the parasitoids and hyperparasitoids found).

Although the response to most variables was similar in both datasets, their significance does change in some cases (Table 2). The abundance of aphids was affected by the structure of the margin vegetation, and was greater where the margin vegetation was predominantly perennial (axis 1) but lower wherever perennial herbaceous species dominated the cover in the field margins (axis 2). The effect of vegetation is consistent in both datasets.

Aphids were significantly less abundant in barley than in wheat fields, with predicted means of 103 aphids/segment vs. 333 aphids/segment (‘time’ dataset) and 47 aphids/segment vs. 161 aphids/segment (‘space’ dataset), respectively. By contrast, aphid abundances were significantly higher in 2014 (predicted mean: 378 aphids/segment) than in both 2015 (predicted mean: 130 aphids/segment) and 2016 (predicted mean: 147 aphids/segment). We detected no differences between geographical areas.

Parasitism rates were also affected by the structure of the field margin vegetation (Table 2) and were significantly positively correlated with a dominance of annual vegetation (axis 1) but negatively correlated with a dominance of perennial herbaceous forms (axis 2). However, the effect of margin vegetation was only significant in the ‘time’ dataset. We detected no significant non-linear relationship. Parasitism rates only had a marginally negative correlation with aphid abundances; we detected no differences between years. However, we did find significant differences between geographical areas that cannot be attributed to the studied variables.

**Table 2 insects-14-00156-t002:** Statistical significance of explanatory variables for aphid abundances and parasitism rates in the ‘time’ and ‘space’ datasets. Model estimates and their significance are shown for the NMDS axis, cereal variety and aphid abundance, while only their overall significance is shown for year and area. Levels of significance: ***: *p*-value < 0.001; **: *p*-value < 0.01; *: *p*-value < 0.05; m: *p*-value < 0.1; n.s.: not significant.

Variable	Aphid Abundances (‘Time’ Dataset)		Aphid Abundances (‘Space’ Dataset)		Parasitism (‘Time’ Dataset)		Parasitism (‘Space’ Dataset)	
axis 1 (linear)	0.477 ± 0.252	*	0.854 ± 0.231	***	−0.334 ± 0.146	*	−0.306 ± 0.254	
axis 1 (quadratic)	0.494 ± 0.312		0.335 ± 0.287		−0.052 ± 0.175		−0.566 ± 0.315	m
axis 2 (linear)	−0.450 ± 0.235	*	−0.535 ± 0.199	**	−0.303 ± 0.125	*	−0.269 ± 0.167	
axis 2 (quadratic)	−0.239 ± 0.371		−0.475 ± 0.319		0.369 ± 0.237		0.233 ± 0.377	
log(aphid abundance)					0.027 ± 0.045		−0.201 ± 0.107	m
barley vs. wheat	−1.173 ± 0.274	***	−1.232 ± 0.267	***				
year		**				ns		
area				ns				*

Ladybug abundances increased with aphid abundances (Table 3). The characteristics of the field margins also affected the abundance of ladybugs and hoverflies as they were significantly more abundant where margins were dominated by annual vegetation (axis 1). However, the abundance of ladybugs did not respond linearly to the structure of the margin vegetation (the quadratic terms were significant for both axes, indicating a hump-shaped relationship). The abundance of ladybugs was lower in areas adjacent to woody margins (axis 2 of the NMDS); the tendency for hoverflies was similar but not significant. The abundances of ladybugs and hoverflies were higher in transects near field margins.

## 4. Discussion

Our results show that margin vegetation can be described using easily obtainable life form traits that not only describe the structure and presence of the vegetation throughout the year but, as in our case, are also related to the resource provisioning (i.e., floral nectar) that is relevant to natural enemies. The life form has the advantage of being very easily determined and can be used therefore as an assessment tool by farmers for field margin management.

Our results also indicate that the vegetation of field margins affects the abundances of pests and their natural enemies, which upholds previously found evidence that field margins play a role in the regulation of both pests and biocontrol agents and their interactions [4,49,50]. Thus, our hypotheses are partially confirmed. More complex vertical structure and more temporal stability, owing to the dominance of woody life forms, correlate with an increased abundance of certain natural enemies.

On the other hand, contrary to our expectations, margins dominated by annual species favored the presence of all studied natural enemies and were correlated with lower aphid abundances in adjacent fields. This pattern suggests that a joint effect of all enemies on aphid populations is taking place and that these annual field margins may promote conservation biological control.

The preference of natural enemies for margins dominated by annual species may be related to a greater availability of sugars and of alternative prey items and hosts on which adult natural enemies of aphids can feed or oviposit. According to our results, these margins tend to have greater cover of entomophilous flowers, a particularly important feeding resource for hoverfly adults, e.g., [4,50]. In other studies, planted flower strips are used in intensive or simplified cereal cropping landscapes to increase the abundance of natural enemies and enhance the biological control of cereal aphids [51,52,53].

Parasitism also responds positively to annual vegetation cover in margins, even though adult parasitoids also use aphid honeydew as a feeding resource. Although honeydew is inferior in terms of nutrition and nectar may thus be preferred [54], honeydew feeding must be considered when evaluating the benefits of greater floral resource availability and increased parasitism rates [55] as it can confound the relationship between nectar presence and parasite abundance.

Another surprising result was that ladybugs also responded to marginal vegetation structure and were present in greater abundances close to margins covered by annual plants. An analysis of the gut content of ladybugs showed that they consume pollen from many plant families such as Gramineae, Compositae, Umbelliferae and Gentianaceae found in field margins [56]. Even though pollen is usually only considered to be an alternative feeding resource for ladybugs during the aphid season [56,57], an aphid-based diet may have to be complemented with pollen [58]. Although all plants produce pollen at some point, only relatively early-flowering habitats such as annual-dominated ruderal field margins may possibly provide a significant source of pollen in late spring, and so ladybugs may respond positively to this type of field margin. Whether or not they use margins for resource provisioning or simply as corridors through which to colonize fields, the importance of margins for ladybugs is underlined by the fact that their abundances are significantly lower further into fields.

The positive effect of margins dominated by annual life forms on ladybugs and parasitoids may also be the result of less pressure from other natural enemies such as carabids and spiders, which are more sensitive to agricultural management and thus more dependent on persistent semi-natural (non-managed) habitats [59,60]. Margins dominated by annual vegetation may also ensure a greater availability of alternative prey items or hosts early in the season if other natural enemies are less abundant, thereby stabilizing populations of ladybugs and parasitoids before cereal aphids increase in the crops and thus enhancing biological controls when most necessary.

The response of natural enemies to the woodiness of the margin vegetation (axis 2 of the NMDS) is more ambiguous: while parasitism rates and hoverfly abundances are higher in areas adjacent to woody margins (although not significantly so in the latter case), the abundance of ladybugs is lower. This response of ladybugs can be interpreted in terms of the negative interactions with other levels of the trophic network that may be taking place. Woody vegetation in margins encourages birds to nest [61] and adults of insectivorous species predate on groups such as ladybugs and hoverflies—and particularly on their larvae—and so disrupt the efficiency of any biological control [62]. The presence of birds in woody margins could explain the negative effect of such margins on ladybug abundances and the lack of significance in the response of both ladybugs and hoverflies to woodiness (axis 2 of the NMDS). Coincidentally, in our data, areas near woody margins hosted higher aphid abundances, which agrees with previous studies reporting that a significant disruption in the biological control increases aphid abundances [62].

Perennially vegetated herbaceous margins are known for their quality as sites for the overwintering and reproduction of ground beetles [63,64]. Some selective-exclusion experiments have found a negative interaction between natural flying and ground-dwelling enemies of aphids such as ground beetles [65], which demonstrates that there is no improvement in biological control if the additive effect of these two guilds is predicted. An example of this negative interaction is the predation by carabid beetles such as *Pterostichus (Morphnosoma) melanarius* (Illiger, 1798) on aphid mummies, which reduces parasitism rates and thus limits the efficiency of biological control [66]. Negative interactions between flying and ground-dwelling natural enemies could be the cause of the lack of significance in the response of ladybugs and hoverflies to increasingly woody margins (neither woody margins with birds nor herbaceous margins with other enemies are optimal), as well as the observed higher parasitism rates in areas adjacent to woody margins (where there is no competition from other enemies and, instead, there are higher abundances of aphids).

## 5. Conclusions

According to our results, different field margins benefit the studied natural enemies, but they have varying value for different groups of arthropods. Whereas margins with a dominance of annual species tend to have a positive effect on all examined groups, perennial margins dominated by woody vegetation seem to favor ladybug presence only. These facts highlight the importance of margin vegetation structure for arthropod communities in cereal fields, and thus for conservation biological control.

By managing vegetation and promoting specific life forms in already existing margins, farmers can enhance the abundance of natural enemies and decrease the pest pressure on their fields. However, in order to guide farmers properly, the effectiveness of biological control and its effects on yields also need to be assessed. Further research should explore temporal trends throughout the growing season and attempt to understand the role that field margin vegetation plays at different times of the year. Additional research on the interaction with other trophic guilds (e.g., birds or ground beetles), as well as the role of secondary nutrients on the vitality of natural enemies could also clarify the potential of field margins for conservation biological control.

## Figures and Tables

**Figure 1 insects-14-00156-f001:**
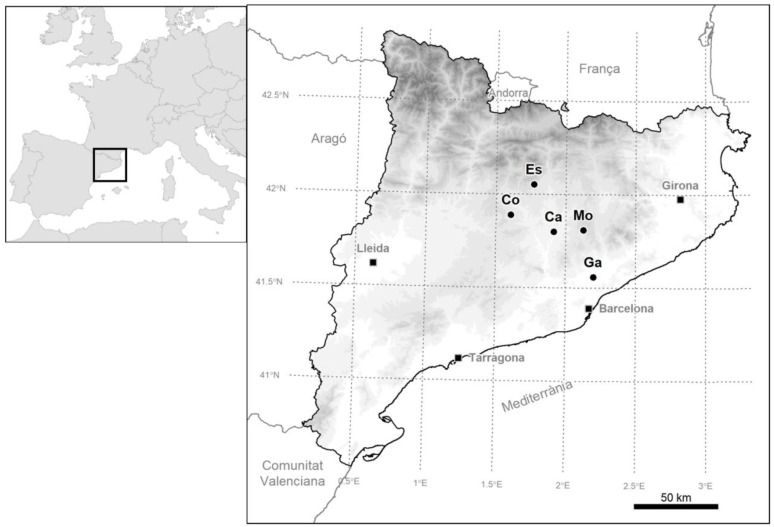
Study areas (circles): Cabrianes (Ca), Cardona (Co), l’Espunyola (Es), Gallecs (Ga) and Moià (Mo). Gray shading indicates elevation every 500 m.

**Figure 2 insects-14-00156-f002:**
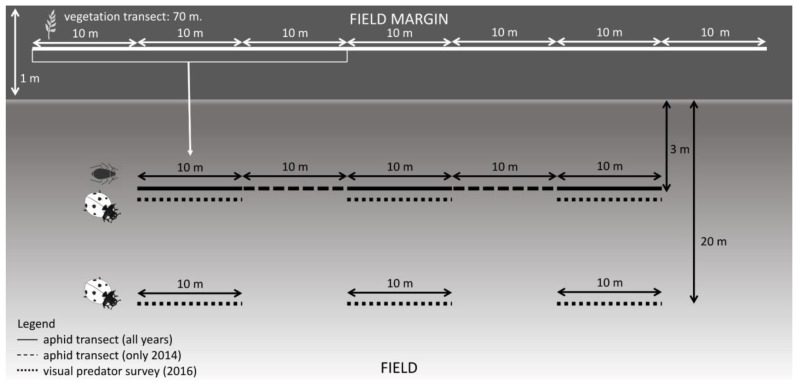
Sampling layout of each of the variables used in the study.

**Figure 3 insects-14-00156-f003:**
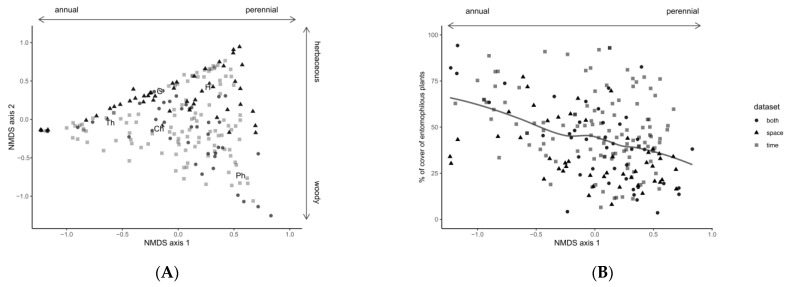
(**A**) Non-metric multidimensional scaling of field margin vegetation; the overlaid labels indicate the centroid of each life form: Th: therophytes; G: geophytes; H: hemicryptophytes; Ch: chamaephytes; Ph: phanerophytes. (**B**) Correlation between the first axis of the margin vegetation NMDS and the entomophilous plant cover. The line indicates the locally estimated smoothing (LOESS) trend for illustrative purposes.

**Table 3 insects-14-00156-t003:** Model estimates and their significance for ladybug and hoverfly abundances. Levels of significance: ***: *p*-value < 0.001; **: *p*-value < 0.01; *: *p*-value < 0.05.

Variable	Ladybug Abundance		Hoverfly Abundance	
axis 1 (linear)	−3.514 ± 0.793	***	−1.429 ± 0.449	**
axis 1 (quadratic)	−2.472 ± 1.228	*	0.282 ± 0.228	
axis 2 (linear)	0.404 ± 0.339		−1.626 ± 0.886	
axis 2 (quadratic)	4.246 ± 1.006	***	0.669 ± 0.595	
20 m vs. 3 m from margin	−0.560 ± 0.280	*	−1.910 ± 0.268	***
log(aphid abundance)	1.297 ± 0.359	***	0.233 ± 0.206	

## Data Availability

The data presented in this study are available on request from the corresponding author.

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
