# Peer review of "Not All Field Margins Are Equally Useful: Effects of the Vegetation Structure of Margins on Cereal Aphids and Their Natural Enemies"

_insects, 2023, doi:10.3390/insects14020156_

Round 1
Reviewer 1 Report
The manuscript titled “Not all field margins are equally useful: effects of the vegetation structure of margins on cereal aphids and their natural enemies” describes the influence of vegetation around areas cultivated with cereal on the occurrence of aphids and natural enemies. The authors found that the abundance of natural enemies predominate on margins dominated by annual plants rather than perennial plants. While aphids predominance occurs on margins dominated by perennial plants rather than annual plants. The introduction, results and discussion sections seemed well written. However, the methodology should be improved. I understood who identified each insect at the species or genus level. There is no mention of how the aphids, parasitoids and hyperparasitoids were identified. Were they identified based on taxonomic aspects? Molecular biology? This kind of information should be addressed to clarify how the work was carried out. Based on my comments, the manuscript needs only minor revisions.
Minor comment
L318-320: The authors mentioned “In other studies” and cited only one reference. Please, correct.
Author Response
Dear reviewer,
Thank you very much for your helpful, and most kind review of manuscript entitled “Not all field margins are equally useful: effects of the vegetation structure of margins on cereal aphids and their natural enemies”.
The manuscript titled “Not all field margins are equally useful: effects of the vegetation structure of margins on cereal aphids and their natural enemies” describes the influence of vegetation around areas cultivated with cereal on the occurrence of aphids and natural enemies. The authors found that the abundance of natural enemies predominate on margins dominated by annual plants rather than perennial plants. While aphids predominance occurs on margins dominated by perennial plants rather than annual plants. The introduction, results and discussion sections seemed well written.
However, the methodology should be improved.
>>>> We have tried to improve it now in this new version
I understood who identified each insect at the species or genus level. There is no mention of how the aphids, parasitoids and hyperparasitoids were identified. Were they identified based on taxonomic aspects? Molecular biology? This kind of information should be addressed to clarify how the work was carried out.
>>> This information is now available in the new version [The aphids were identified by Nicolás Pérez Hidalgo, the parasitoids (Braconidae: Aphidiinae) by José M. Michelena Saval, and the hyperparasitoids (Hymenoptera), by Mar Ferrer Suay (Cynipoidea: Figitidae: Charipinae), Emilio Guerrieri (Chalcidoidea: Encyrtidae) and Agnès Salat Moltó (Chalcidoidea: Pteromalidae, Aphelinidae; Ceraphronoidea: Megaspilidae).]. We have specified now that all identification were based on morphology-based taxonomy and relevant references are included
Minor comment
L318-320: The authors mentioned “In other studies” and cited only one reference. Please, correct.
>>>That’s right! Another reference has been added (please check line 449)
Reviewer 2 Report
Overall the study is well conducted, data are analyzed appropriately and the manuscript fairly written. The lack of graphs makes the paper less appealing. In the paper is also not stressed at all the concept of “conservation biological control”, where the filed margins play a key role, see for example:
Holland, J. M., Bianchi, F. J., Entling, M. H., Moonen, A. C., Smith, B. M., & Jeanneret, P. (2016). Structure, function and management of semi‐natural habitats for conservation biological control: a review of European studies. Pest management science, 72(9), 1638-1651.
Balzan, M. V., & Moonen, A. C. (2014). Field margin vegetation enhances biological control and crop damage suppression from multiple pests in organic tomato fields. Entomologia Experimentalis et Applicata, 150(1), 45-65.
Cusumano, A., Bella, P., Peri, E., Rostás, M., Guarino, S., Lievens, B., & Colazza, S. (2022). Nectar-inhabiting bacteria affect olfactory responses of an insect parasitoid by altering nectar odors. Microbial Ecology, 1-13.
Some specific comments:
In the section “Sampling site and field selection” the author should be more accurate in describing the time when the samplings were carried out, not just “spring” is too generic.
In the section “Insect data analyses”, there is a repetition too many time of sentences starting with “we used… we employed.. etc” try to make it smoother
Lines 203, missing dot
Lines 296-298, something is missing in the sentence
Lines 299-302, unclear sentence.
Lines 365-368, cumbersome sentence
Author Response
Dear reviewer,
Thank you very much for your helpful, and most kind review of manuscript entitled “Not all field margins are equally useful: effects of the vegetation structure of margins on cereal aphids and their natural enemies”.
Overall the study is well conducted, data are analyzed appropriately and the manuscript fairly written. The lack of graphs makes the paper less appealing.
>>>> Thanks for the comment. In fact the Figures were uploaded but unfortunately were not included in the manuscript; we have now included the Figures in the manuscript.
In the paper is also not stressed at all the concept of “conservation biological control”, where the filed margins play a key role, see for example:
Holland, J. M., Bianchi, F. J., Entling, M. H., Moonen, A. C., Smith, B. M., & Jeanneret, P. (2016). Structure, function and management of semi‐natural habitats for conservation biological control: a review of European studies. Pest management science, 72(9), 1638-1651.
Balzan, M. V., & Moonen, A. C. (2014). Field margin vegetation enhances biological control and crop damage suppression from multiple pests in organic tomato fields. Entomologia Experimentalis et Applicata, 150(1), 45-65.
Cusumano, A., Bella, P., Peri, E., Rostás, M., Guarino, S., Lievens, B., & Colazza, S. (2022). Nectar-inhabiting bacteria affect olfactory responses of an insect parasitoid by altering nectar odors. Microbial Ecology, 1-13.
>>>> We completely agree, this omission is solved and conservation biological control term is now included and also the suggested references.
Some specific comments:
In the section “Sampling site and field selection” the author should be more accurate in describing the time when the samplings were carried out, not just “spring” is too generic.
>>> That’s right. In relation to aphids samplings we offered all details, because we sampled once per week (every Monday to be precise) but we have included now more details also for the sampling of natural enemies.
In the section “Insect data analyses”, there is a repetition too many time of sentences starting with “we used… we employed.. etc” try to make it smoother
>>>> We agree, we have tried to improve it now with a new version with the English revised
Lines 203, missing dot
>>>Corrected!
Lines 296-298, something is missing in the sentence
>> Sure! Sentence rephrased (please check line 431-435)
Lines 299-302, unclear sentence.
>> We agree! We have changed it (please read line 436-440)
Lines 365-368, cumbersome sentence
>>> We see your point. We have changed the first sentence of conclusions
According your selection in the Table:
>>> We have rephrased the conclusions in order to improve the final message
>>> We hope that with all Figures attached the results of this new version will be clearer
Again. thank you very much for your helpful, and most kind review

Reviewer 3 Report
Dear Autors,
I found the article very good and read it with great interest. We need more studies, considering further levels of plant-insect interactions and “disentangle food chains” in the field margins. Nevertheless, I have some stylistic and content-related comments. The latter can also be seen as recommendations for follow-up studies.
Simply summary: here I would avoid “personal” addresses and just write: Which plant life forms are dominant in the field margin vegetation…
Line 22 – sampled aphids as well as some of their natural enemies
The introduction can be a bit improved. E.g. – you concentrate on the plant life form and hardly mention other functional traits, which could be also relevant. Later on, in the discussion, you also refer to the different kinds of reward (nectar vs pollen) and the phenology of plant species. These are either important traits, which should be already mentioned in the introduction. By the way, a rough concept of life forms sensu Raunkiær can be improved for your purposes by distinguishing additional categories suchlike: between winter and summer annuals; graminoid (wind-pollinated) and herbaceous hemikryptophytes and geophytes; rosulate, semi-rosulate and erosulate plants (quite a good predictor for leaf canopy height at herbaceous species).
The first sentence on line 61 is not connected to the following paragraph and can be omitted.
Line 64 – Do you mean “even if their species composition differ”?
Line 67 – From the many aspects…
Lines 69-70 this sentence should be reformulated - you probably mean “permanent vegetation cover”
Line 75-76 This sentence should be removed towards the next paragraph. In this way, you separate the part of the Introduction dealing with the current state of research and the part of the Introduction, which can be called “Questions and Hypotheses”
Line 92 – better – “cereal crops during the growth season…
Methods are well-described and comprehensible.
Line103 – by mentioning “perennial grasslands” you should be aware that it is a very broad concept, ranging from very species-poor graminoid-dominated loans to species and especially flower-rich strips. Maybe a short sentence about what category prevails under what you count to perennial grasslands should be added.
Line 105 “data collected in the area”
Lines 140-144 – please add citations for literature or reference collections/databases used for the species identification (if any).
Line 181 Community weighted means (plural)
Line 198 - two independent datasets for “Time” and “Space” dataset.
Results
I would recommend adding one more figure (graph) visualizing more detailed the abundance of different groups of insects (ladybugs, hoverflies) in NMDS (Figure 3 is also important, but a bit difficult to grasp and does not provide information concerning both mentioned groups). Use e.g. the different sizes of the dots to illustrate the abundance of ladybugs/hoverflies per sample site.
Line 232 (proposal) “Most of them dry up in summer and leave patches of bare soil behind. In contrast, margins dominated by perennials are vegetated throughout the year.”
Line 235 - “plants in bloom during the sampling period
Line 241 - “Unlike hemicryptophytes, phanerophytes and therophytes, whose percentages of cover vary considerably from 0% to almost 100%,” – what time period is meant for these fluctuations? The whole year or only your sampling time between April and June? Please be precise.
Discussion:
Discussing the role of entomophilous flowers, you should notice, that there is an important differentiation not only between the annual weeds and perennials / woody species but e.g. between plants belonging to different phylogenetic groups. The quality of nectar and/or pollen as food for insects can differ significantly between families. For solitary bee species (Apoidea) is known, that some species only feed on flowers of a particular plant family. You even mention some plant families (Lines 330-331), which can be relevant for ladybirds. Are any preferences known for hoverflies?
All in all, a short statement about the relevance of other plant functional traits (besides live form) could be added (see my remark to the introduction)
The second part of the discussion is well-written and there were several points that sound very interesting for me. E.g. the effects of the habitat structure and interactions with further organism groups like birds and ground beetles. The same is true for the statement, that insects using aphids or aphid honey drew as the main nutrition could still be dependent on additional food provided by plants (nectar, pollen). Do some additional substances such as vitamins or microelements play a role and does their absence in the diet of imagines of parasitoids or ladybirds reduce their vitality? Are there any explanations for it in the literature? All these could be important research questions for the future.
The conclusions are consistent and correspond to the main findings.
Author Response
Dear reviewer,
Thank you very much for your helpful, and most kind review of manuscript entitled “Not all field margins are equally useful: effects of the vegetation structure of margins on cereal aphids and their natural enemies”.
Dear Autors,
I found the article very good and read it with great interest. We need more studies, considering further levels of plant-insect interactions and “disentangle food chains” in the field margins. Nevertheless, I have some stylistic and content-related comments. The latter can also be seen as recommendations for follow-up studies.
Simply summary: here I would avoid “personal” addresses and just write: Which plant life forms are dominant in the field margin vegetation…
>>Done
Line 22 – sampled aphids as well as some of their natural enemies
>>Done
The introduction can be a bit improved. E.g. – you concentrate on the plant life form and hardly mention other functional traits, which could be also relevant. Later on, in the discussion, you also refer to the different kinds of reward (nectar vs pollen) and the phenology of plant species. These are either important traits, which should be already mentioned in the introduction.
>>Thanks for your comment, we see your point. We have tried to improve it. Please read lines 112-116
By the way, a rough concept of life forms sensu Raunkiær can be improved for your purposes by distinguishing additional categories suchlike: between winter and summer annuals; graminoid (wind-pollinated) and herbaceous hemikryptophytes and geophytes; rosulate, semi-rosulate and erosulate plants (quite a good predictor for leaf canopy height at herbaceous species).
>>>>Thank you for these suggestions. It would indeed be interesting analyses to carry out with our dataset to further clarify the effect of margin vegetation. We have included this on the further research section, please see Conclusions.
The first sentence on line 61 is not connected to the following paragraph and can be omitted.
>>Done
Line 64 – Do you mean “even if their species composition differ”?
>>>Yes, corrected
Line 67 – From the many aspects…
>>>Done
Lines 69-70 this sentence should be reformulated - you probably mean “permanent vegetation cover”
>>>This sentence has been modified in the English revision, please read line 108
Line 75-76 This sentence should be removed towards the next paragraph. In this way, you separate the part of the Introduction dealing with the current state of research and the part of the Introduction, which can be called “Questions and Hypotheses”
>>>We have divided the paragraph and tried to restructure the section (please read 126-135).
Line 92 – better – “cereal crops during the growth season…
>>>Sure. Done
Methods are well-described and comprehensible.
Line103 – by mentioning “perennial grasslands” you should be aware that it is a very broad concept, ranging from very species-poor graminoid-dominated loans to species and especially flower-rich strips. Maybe a short sentence about what category prevails under what you count to perennial grasslands should be added.
>>>>> Perennial grasslands in the study area are dominated by perennial grasses, and are relatively poor in species and flower resources. We have specified so (please read lines 154-156)
Line 105 “data collected in the area”
>> Corrected in the new version
Lines 140-144 – please add citations for literature or reference collections/databases used for the species identification (if any).
>> A really pertinent comment. References added in the new version
Line 181 Community weighted means (plural)
>>>Done
Line 198 - two independent datasets for “Time” and “Space” dataset.
>>>Done
Results
I would recommend adding one more figure (graph) visualizing more detailed the abundance of different groups of insects (ladybugs, hoverflies) in NMDS (Figure 3 is also important, but a bit difficult to grasp and does not provide information concerning both mentioned groups). Use e.g. the different sizes of the dots to illustrate the abundance of ladybugs/hoverflies per sample site.
>>> We have worked on a figure to display the requested information, but it is not particularly clear (neither with dots of different sizes, nor with a smoothed response surface). We can include, however, some figures displaying the overall trends of the fitted models (please pdf attacched)
Line 232 (proposal) “Most of them dry up in summer and leave patches of bare soil behind. In contrast, margins dominated by perennials are vegetated throughout the year.”
>>> This paragraph has been rephrased in the new version, although not exactly as proposed because not all these non-perennial margins dry up in summer. Please read line 328-330
Line 235 - “plants in bloom during the sampling period
>>> Sentence corrected in the new version. Please read line 323
Line 241 - “Unlike hemicryptophytes, phanerophytes and therophytes, whose percentages of cover vary considerably from 0% to almost 100%,” – what time period is meant for these fluctuations? The whole year or only your sampling time between April and June? Please be precise.
>>> This variation is not across time, but across the diversity of field margins (it means that geophytes and chamaephytes have equally low cover in all the sampled field margins, and therefore have very little discriminant power).
Discussion:
Discussing the role of entomophilous flowers, you should notice, that there is an important differentiation not only between the annual weeds and perennials / woody species but e.g. between plants belonging to different phylogenetic groups. The quality of nectar and/or pollen as food for insects can differ significantly between families. For solitary bee species (Apoidea) is known, that some species only feed on flowers of a particular plant family. You even mention some plant families (Lines 330-331), which can be relevant for ladybirds. Are any preferences known for hoverflies?
>>>>>Some hoverflies have also been shown to exhibit innate colour preferences Similarly, some authors have found that certain species of plants may help attracting various species of aphidophagous hoverflies. However, since hoverfly abundance was based on larvae presence and species identification was not possible, we thought these preferences were a degree of detail beyond the scope of our study.
All in all, a short statement about the relevance of other plant functional traits (besides live form) could be added (see my remark to the introduction)
>>>>We have tried to improve this issue in this new version
The second part of the discussion is well-written and there were several points that sound very interesting for me. E.g. the effects of the habitat structure and interactions with further organism groups like birds and ground beetles. The same is true for the statement, that insects using aphids or aphid honey drew as the main nutrition could still be dependent on additional food provided by plants (nectar, pollen). Do some additional substances such as vitamins or microelements play a role and does their absence in the diet of imagines of parasitoids or ladybirds reduce their vitality? Are there any explanations for it in the literature? All these could be important research questions for the future.
>>> Thank you! These suggestions have been added to the conclusions as possible further research, as they seem really interesting because we didn’t find studies dealing with it.
The conclusions are consistent and correspond to the main findings.
Again, thanks for your helpful, and most kind review
Yours faithfully,
Berta Caballero López

Reviewer 4 Report
Dear Authors, I appreciate your paper.
I suggest to describe the fields you choosed, are they contiguous, are similarly managed, which kind of borders they have. Why you collected data in GA for three years (2014-2016) and in the other three only in 2015? In GA weather conditions were similar in the three years?

Author Response
Dear reviewer,
Thank you very much for your helpful, and most kind review of manuscript entitled “Not all field margins are equally useful: effects of the vegetation structure of margins on cereal aphids and their natural enemies”.
Dear Authors, I appreciate your paper.
I suggest to describe the fields you choosed, are they contiguous, are similarly managed, which kind of borders they have.
>>>>>We have included some information about fields studied and how they were managed, please read lines 153-158
Why you collected data in GA for three years (2014-2016) and in the other three only in 2015?
>>>>Aphid populations show great variability from one year to another and from one place to another. As specified in lines 168-170 we were interested in the consistency of the response of both aphid populations and their enemies in time and in space, hence the difference in data collection. Ideally, all areas would have been sampled for three years, but this effort was not possible due to budget constraints.
In GA weather conditions were similar in the three years?
>>>>>We have compiled the meteorological data obtained by the Servei Meteorològic de Catalunya from the closest meteorological stations to the sampling locations between 2014-2016 in Gallecs, and Cabrianes, L’Espunyola, Cardona and Moià in spring 2015. Some authors have found that annual changes in aphid abundances are often related to temperature and precipitation (Leslie et al., 2009; Sampaio et al., 2017; Thies et al., 2005; Vialatte et al., 2007), with lower abundances found in years with extreme weather events. In our case, spring 2015 was extremely hot and dry, while early spring 2016 was characterised by episodes of heavy rain. In both years, aphid abundances were lower than in 2014, when no such extreme events occurred. However, these effects are not the primary interest of our research, and are absorbed by the ‘year’ or the ‘area’ effect in the fitted models, and therefore they are not dealt with in the manuscript.
Where is the Fig.1?
>>>>>We appreciate your first comment because we hadn’t realized that figures should had been included in the body of the text; Figures now have been added in the manuscript
Coccinella septempunctata in italics and the species name in lowercase
>>>>Done
From your comment about Table S1. “Sipha flava to Sitobion avenae: I suggest to alphabetically reorder”
>>>> That’s right. New table alphabetically organized
Thank you very much for your most kind review
Yours faithfully,
